# Automatic planning of liver tumor thermal ablation using deep reinforcement learning

**Krishna Chaitanya**[*1,2]                                        KRISHNA.CHAITANYA@VISION.EE.ETHZ.CH
**Chloé Audigier**[*2]                                          CHLOE.AUDIGIER@SIEMENS-HEALTHINEERS.COM
**Laura Elena Balascuta**[3]                                       LAURA.BALASCUTA@SIEMENS.COM
**Tommaso Mansi**[4]                                                THOMAS.MANSI@GMAIL.COM

[1] *Computer Vision Lab, ETH Zurich, Switzerland.*

[2] *Advanced Clinical Imaging Technology, Siemens Healthcare, Lausanne, Switzerland.*

[3] *Siemens SRL, Brasov, Romania.*

[4] *Siemens Healthineers, Digital Technology & Innovation, Princeton, NJ, USA.*

**Editors:** Under Review for MIDL 2022

## Abstract

Thermal ablation is a promising minimally invasive intervention to treat liver tumors. It requires a meticulous planning phase where the electrode trajectory from the skin surface to the tumor inside the liver as well as the ablation protocol are defined to reach a complete tumor ablation while considering multiple clinical constraints such as avoiding too much damage to healthy tissue. The planning is usually done manually based on 2D views of pre-operative CT images and can be extremely challenging for large or irregularly shaped tumors. Conventional optimization methods have been proposed to automate this complex task, but they suffer from high computation time. To alleviate this drawback, we propose to leverage a deep reinforcement learning (DRL) approach to find the optimal electrode trajectory that satisfies all the clinical constraints and does not require any labels in training. Here, we define a custom environment as the 3D mask with tumor, surrounding organs, skin labels along with an electrode line and ablation zone. An agent, represented by a neural network, interacts with the custom environment by displacing the electrode and therefore can learn an optimal policy. The reward assignment is done based on the clinical constraints. We explore discrete and continuous action-based approaches with double deep Q networks and proximal policy optimization (PPO), respectively. We perform an evaluation on the publicly available liver tumor segmentation (LITs) challenge dataset and obtain solutions that satisfy all clinical constraints comparable to the conventional method. The DRL method does not need any post-processing steps, allowing a mean inference time of 13.3 seconds per subject compared to the conventional optimization method's mean time of 135 seconds. Moreover, the best DRL method (PPO) yields a valid solution irrespective of the tumor location within the liver that demonstrates its robustness.

**Keywords:** Deep reinforcement learning (DRL), liver tumor ablation, thermal ablation, proximal policy optimization (PPO), double deep Q networks (DDQN).

## 1. Introduction

Liver cancer is the fourth highest occurring cancer type (Naghavi et al., 2017) in the world. It can be treated invasively with resection or transplantation or minimally invasively with

---

[*] Contributed equally. This work was done as a part of an internship at Siemens Healthineers.

thermal ablation. For example, radiofrequency ablation is a widely used technique (Garrean et al., 2008; Minami and Kudo, 2011), where a thin electrode is inserted from the skin surface into the tumor. A high-frequency electric field is induced to increase the temperature and generate an ablation in the targeted tumor and surrounding tissues. Thermal ablation requires a meticulous planning phase whose goal is to find the optimal electrode trajectory and ablation protocol that achieves a complete tumor ablation and satisfy certain clinical constraints such as avoiding too much damage to healthy tissue.

Typically, clinicians plan the intervention manually by visualizing the CT images in 2D. Such visual planning is time-consuming and challenging. It can lead to incomplete tumor ablation and the acquisition of many CT images to avoid collision of the electrodes with surrounding organs at risk (OAR). Therefore, computed assisted planning is valuable with the initial solutions offering better visualization for interactive planning tools (McCreedy et al., 2006; Rieder et al., 2009; Khlebnikov et al., 2011; Kerbl et al., 2012).

Conventional optimization-based methods have been proposed using downhill simplex optimization (Baegert et al., 2007a,b), a gradient descent method (Altrogge et al., 2006), Pareto optimality (Seitel et al., 2011; Schumann et al., 2015) and others (Baegert et al., 2007a,b; Schumann et al., 2010; Seitel et al., 2011; Schumann et al., 2015). In a relevant work (Liang et al., 2019), the authors propose to leverage a cover-set-based method to find the set of Pareto optimal electrode trajectories satisfying all clinical constraints. Here, (a) first they identify a set of target tumor points, (b) next, determine all the available entry skin points and select a subset of valid points that satisfy clinical constraints, (c) compute a score based on the clinical constraints for trajectory paths from all the tumor points to all valid skin points, and (d) lastly, these paths and scores information are input to an optimization framework that finds the optimal trajectory paths. Such approaches are computationally expensive since they involve many processing steps with a high inference time per subject ranging from hours to minutes for the fastest method (Liang et al., 2019).

Alternatively, data-driven approaches, especially deep learning (DL) methods, are good at mitigating such inference time issues and yield accurate and quick results. Fully supervised DL approaches have been proposed for automatic planning of medical interventions (Tschannen et al., 2016; Esfandiari et al., 2018; Kulyk et al., 2018; Vercauteren et al., 2019). But they require a large set of annotations for training to yield high performance, and acquiring such electrode trajectories annotations from clinical experts is time-consuming and expensive. Deep reinforcement learning (DRL) based solutions (Sutton et al., 2000; Mnih et al., 2013) can be a promising alternative to supervised approaches as they do not require labeled datasets as explored in (Kober et al., 2013; Ghesu et al., 2017; Krebs et al., 2017; Zhang et al., 2018; Yu et al., 2019). Recently, a DRL-based approach (Ackermann et al., 2021) was proposed for surgery planning of orthopedic hip disorders with training done in a custom simulated environment and later tested on a small dataset of 8 patients. The validation size may not be adequate to deploy for clinical applications. Moreover, training with simulated data and testing on real-world data has been shown to not generalize well (Pan et al., 2017) due to the changes in data distribution. Also, DRL approaches do not involve any processing steps and can generalize well at test-time due to the nature of training.

In this work, to address the above limitations, we present a DRL-based approach with low inference time and no manual annotations requirement during training to find an optimal ablation plan achieving 100% tumor coverage and satisfying all clinical constraints.

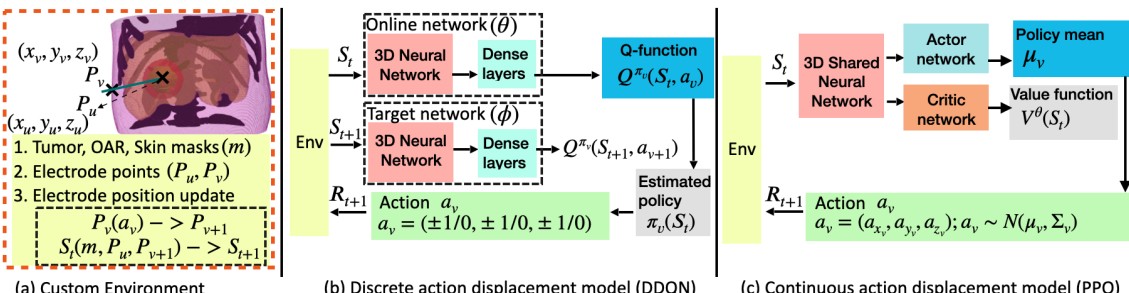

**Figure 1:** (a) Custom environment defined with organs at risk, tumor, skin masks, and electrode trajectory with ablation zone. (b) discrete action based displacement of skin electrode endpoint done using a double DQN approach. (c) continuous action based displacement of skin electrode endpoint done with a proximal policy optimization method.

Specifically, we explore two approaches: (a) in the first one, we explore a discrete displacement of electrode endpoint with a double deep Q networks (Van Hasselt et al., 2016) (DDQN) that estimates discrete action values. We also evaluate hindsight experience replay (Andrychowicz et al., 2017) (HER) with DDQN. (b) in the second, we explore the continuous displacement of electrode endpoint with a policy gradient method, proximal policy optimization (Schulman et al., 2017) (PPO) that outputs the continuous action policy directly. We evaluate both approaches on a public dataset (LITs) (Bilic et al., 2019) and obtain results comparable to a simplified conventional optimisation implementation based on (Liang et al., 2019). We perform additional analysis to evaluate the robustness of the approach with respect to the tumor location within the 9 liver segments (refer to Fig. 3).

## 2. Methods

To automate the tedious liver tumor thermal ablation planning, we leverage a deep reinforcement learning (DRL) approach to predict the optimal electrode trajectory satisfying all the clinical constraints. Reinforcement learning is a framework where an agent learns how to interact with an environment based on experiences. The objective is to maximize the cumulative rewards by learning an optimal policy that gives a set of actions to take from a current state to reach the terminal state. The environment state is updated based on the agent's action and a reward is obtained. This iteration is done till the final state is reached.

### 2.1. Custom Environment (Env)

In Fig. 1.a., we present the custom environment consisting of the mask m, containing 3D information of the tumor, organs at risk and skin masks, and the electrode endpoints $(P_u, P_v)$ with the ablation zone. The state $S_t$ is defined based on this combined information as $S_t = (m, P_u, P_v)$. The electrode tumor endpoint $P_u$ is fixed at the center of the tumor. Here, the ablation zone is modeled as a sphere centered at $P_u$ whose radius is chosen from a given set of valid radius values to achieve a 100% tumor coverage. The electrode skin endpoint $P_v = (x_v, y_v, z_v)$ is randomly assigned outside the skin surface while ensuring an electrode length less than 150mm (clinical constraint).

| Clinical Constraints | Rewards for skin endpoint $P_v$ |
|---|---|
| 1. Electrode trajectory must not collide with OAR | $+1$ |
| 2. Electrode skin endpoint $P_v$ must be outside the body | $+1$ |
| 3. Length of electrode within the body ($l_e$) is less than the maximum allowed electrode length (150 mm) | $r_l = (150 - l_e)/150$ |
| 4. The distance ($d_{oar}$) between OAR and the electrode is at least 12 mm | $r_d = d_{oar}/100$ |
| 5. Ablation zone on $P_u$ must have 100% tumor coverage | NA |
| 6. Ablation zone must not have any collision with OAR | NA |

**Table 1:** Clinical constraints and corresponding rewards for skin electrode endpoint $P_v$.

In a given state $S_t$ (**initial state** for each subject), the network outputs an action $a_v$ which is used to update $P_v$ to $P_{v+1}$. The next state $S_{t+1}$ is computed using the updated electrode endpoints ($P_u, P_{v+1}$) and the mask $m$. This is repeated until either the terminal state is reached or a maximum number of 50 steps is reached. The net reward $R_{t+1}$ is estimated based on the updated state $S_{t+1}$ with new endpoint. Based on a pre-defined net reward value, we determine whether to continue to move to a new electrode state or terminate. The termination happens when it satisfies all the clinical constraints (Table 1) and is called a **terminal or final state**, where the reward value is 2.12 or greater.

**Clinical constraints and rewards:** The clinical constraints and corresponding rewards are given in Table 1. For constraint 3, $r_l$ is positive when the electrode length is less than 150 mm, and negative when it is greater than 150 mm. In terminal state, all clinical constraints must be satisfied, so, the minimum reward is 2.12 with $r_d \geq 0.12$ and $r_l > 0$.

We explored two approaches: a double deep Q networks (Sec. 2.2) where the electrode endpoints displacement are modeled using a discrete action space and a proximal policy optimization (Sec. 2.3) where a continuous action space is used.

## 2.2. Double deep Q networks (DDQN)

To model the displacement of electrode endpoint in discrete steps, we use a value learning approach, the double deep Q network (DDQN) (Van Hasselt et al., 2016). The outline of this approach is presented in Fig. 1.b. Here, we have a pre-defined number of possible actions. Given an input state $S_t$, the online network estimates the Q-values $Q^{\pi_v}(S_t, a_v)$ for all the pre-defined actions. The action with the highest Q-value is used to estimate action policy $\pi_v(S_t)$ and chosen as the best action $a_v$ that is later applied to get the updated state $S_{t+1}$ and corresponding reward $R_{t+1}$. The discrete action values are $a_v = (\pm 1/0, \pm 1/0, \pm 1/0)$ meaning that each coordinate of the electrode endpoint can be updated by $+1$, $-1$ or $0$ (no displacement) in the 3D space. Thereby, the dense layers output 27 ($3^3$) Q-values corresponding to the combinations of possible actions. The skin electrode endpoint is updated as $P_{v+1} = P_v(a_v) = (x_v \pm 1/0, y_v \pm 1/0, z_v \pm 1/0)$.

We have two sets of networks as proposed in DDQN (Van Hasselt et al., 2016), shown in Fig. 1.b., referred to as online and target networks with parameters $\theta$ and $\phi$, respectively. The online network aims to reach the Q-value estimated by the target network by the end of training. The online network weights are updated by optimizing the mean squared error loss as defined in Eqn. 1 and the target network weights are updated periodically with the

online network weights $\phi = \theta$ after every $N$ number of training episodes.

$$L_d = \mathbb{E}[\| (R_t + \gamma Q^{\pi_v}(S_{t+1}, \pi_v(S_{t+1}); \phi)) - Q(S_t, a_v; \theta) \|^2] \tag{1}$$

Here, the first term is the target network Q-value estimate, and the second term is the online network Q-value estimate. In the first term, $\gamma$ denotes the discount factor used in the cumulative reward estimation.

**Hindsight experience replay (HER)**: Since the electrode trajectory solutions exist sparsely, we explore HER used with DDQN that has provided performance gains for such sparse reward problems (Andrychowicz et al., 2017). For HER, we consider final states that do not reach the terminal state as an additional "terminal" state if they satisfy the clinical constraints of $1, 2, 3$ and do not satisfy the constraint 4 (min. distance to OAR $> 12mm$). In those cases, this distance would lie between 0 and 12mm as enforced by constraint 1.

## 2.3. Proximal Policy Optimization

For the continuous displacement of electrode endpoint, we consider a popular policy gradient method called Proximal Policy Optimization (PPO), whose outline is presented in Fig. 1.c. With this approach, a shared 3D network followed by two smaller dense layer networks called actor and critic networks are defined. The actor-network estimates the mean $\mu_v$ of the action policy while the critic network estimates the value-function $V^\theta(S_t)$ of the state ($\theta$ denotes the network parameters). $\mu_v$ is multi-dimensional (3 values for the 3D coordinates of $P_v$). The action policy $\pi_\theta(a_v|S_t)$ is defined as a multivariate Gaussian distribution $N(\mu_v, \Sigma_v)$ with the mean value $\mu_v$ output from the network and a fixed variance value $\Sigma_v$. Next, a random continuous action value $a_v = (a_{x_v}, a_{y_v}, a_{z_v})$ is sampled from the Gaussian distribution $N(\mu_v, \Sigma_v)$ and applied to get updated electrode skin endpoint: $P_{v+1} = P_v(a_v) = (x_v + a_{x_v}, y_v + a_{y_v}, z_v + a_{z_v})$.

The net loss for the PPO approach training is defined as below:

$$L_c = \mathbb{E}_t \left[ L_{clip}(\theta) - c_1 L_t^{VF}(\theta) + c_2 S[\pi_v](S_t) \right] \tag{2}$$

The first term is the clipped loss: $L_{clip}(\theta) = \mathbb{E}_t \left[ min(r_t(\theta)\hat{A}_t, clip(r_t(\theta), 1 - \varepsilon, 1 + \varepsilon)\hat{A}_t) \right]$ Here, $\varepsilon$ controls the change in policy, $r_t$ is the ratio of likelihood of actions under current vs old policy defined as $r_t(\theta) = \frac{\pi_\theta(a_t|s_t)}{\pi_{\theta_{old}}(a_t|s_t)}$. $\hat{A}_t$ is the advantage function that measures the relative positive or negative reward value for current set of actions w.r.t an average set of actions. It is defined as: $\hat{A}_t = \hat{R}_t - V^\theta(S_t)$, where $\hat{R}_t$ is the cumulative rewards given by $\hat{R}_t = r_t + \gamma * r_{t+1} + .. + \gamma^{T-t}V^\theta(S_t)$, $T$ is maximum number of steps allowed in an episode.

The second term is the mean squared error of value function: $L_t^{VF}(\theta) = \|\hat{R}_t - V^\theta(S_t)\|^2$. The hyper-parameter $c_1$ controls the contribution of this loss term from the critic network.

The third term is the entropy term that dictates policy exploration with the hyper-parameter $c_2$ where a lower $c_2$ value means lower exploration, and vice-versa.

## 3. Datasets and training details

**Dataset:** The LITs dataset (Bilic et al., 2019) is a publicly available dataset that contains 130 CT scans with expert annotations for tumors and livers. Each CT volume contains

multiple tumors. We select a maximum of 10 tumors per subject that can be ablated without collision with any OAR leading to a total of 496 cases, which we split into training, validation, and test sets containing 225, 131, and 140 cases, respectively.

**Pre-processing:** First, the segmentation of the organs at risk (OAR), and 9 segments of the liver are generated automatically using a deep learning image-to-image network (Yang et al., 2017). The OAR mask contains spleen, heart, right and left kidney, right and left lung, bladder, spinal cord, aorta, aorta hepa, ribs, skeleton, hepatic vessels and portal veins. Then, we define a combined 3D mask $(m)$ with tumor, liver, OAR, and skin labels by applying the below steps sequentially: (1) dilation of 1 mm to ribs, skeleton, blood vessels in the liver, and 5mm to other OAR. (2) compute the ablation sphere radius for the tumor at $1mm^3$ resolution. (3) re-sample mask $(m)$ at $3mm^3$. (4) crop the mask based on the liver segmentation to reduce its dimensions and remove the unrealistic entry skin points located on the patient back. (5) compute the distance map to OAR, excluding blood vessels in the liver. Finally, crop the mask and distance map to fixed dimensions of (96,90,128).

**Network Architecture:** The 3D network has the same architecture for both DDQN and PPO approaches. It has three 3D convolution layers with filter, kernel size, and strides of (32,8,4), (64,4,2), (64,3,1), respectively. The resultant output is flattened and passed through a dense layer with 512 units output. All the above layers have ReLU activations. For DDQN, we have a dense layers network that takes these 512-units as input and returns 27 values corresponding to Q-values. For PPO, we have two outputs from actor and critic networks following the shared network. The actor-network has two dense layers with first a dense layer of 64 outputs, followed by ReLU, and lastly with a dense layer of 3 output values (mean values). Similarly, the critic-network has two layers with a dense layer of 64 outputs, followed by ReLU, finally with a final dense layer of 1 output value (value estimate).

**Training Details:**

**DDQN:** In each episode, we sample a random subject and try to reach the terminal state within a maximum of 50 steps by either exploration (randomly sampled action out of all possible actions) or exploitation (optimal action predicted by online network). All these experiences are populated in an experience replay buffer (Zhang and Sutton, 2017) that stores all the (state, action, next state, reward) pairs in the memory referred to as experiences. At the start, we explore more and accumulate experiences. After reaching a pre-defined number of experiences, in each episode, the online network is trained on a batch of randomly sampled experiences from the replay buffer with the loss given in Eqn. 1. Here, batch size was set to 32 and learning rate to $5e^{-4}$. We evaluated five values of $\gamma$: $0.1, 0.2, 0.3, 0.4, 0.5$. The exploration and exploitation are controlled by a variable $\epsilon$ initially set as 1, which decays with a decay rate of 0.9995. At the start of training, more exploration is done while towards the end, more exploitation is used. The target network weights $\phi$ are updated with the network weights of the online network $\theta$ periodically every 10 episodes. We found that training the networks for 2000 episodes led to a stable convergence.

**PPO:** In each episode, we sample a random subject. Next, the skin endpoint $P_v$ is displaced to reach the terminal state within 50 steps. The network is updated at the end of the episode with the loss based on this episode's steps. In each episode training, we perform a joint optimization with both the first and second loss terms in Eqn. 2 as we did not observe any performance gains using the third term of entropy loss and set $c_2$ to 0. We train the network for 2000 episodes with a learning rate of $5e^{-4}$. The hyper-parameters

$c_1$, $\Sigma_v$, $\varepsilon$ (ppo clip value) are empirically set to 1, 0.5, and 0.2, respectively. As suggested in (Schulman et al., 2015, 2017), in an episode, we stop the network updates when the mean KL divergence estimate (ratio of previous log probabilities to new log probabilities) for a given training example exceeds a threshold value, set to 0.02.

The **training time** for DRL methods DDQN, DDQN+HER, and PPO were 9.4, 7.2, and 10 hours, respectively. All models were trained on Quadro RTX 8000 GPU.

**Evaluation:** For each test subject, we consider 10 random initialization of the electrode skin endpoint $P_v$. The corresponding state $S_t$ is passed through the trained network which either reaches a valid solution (terminal state, satisfying all constraints) within 50 steps or not. If a valid solution is found, the accuracy is set to 1. Else, it is set to 0 and declared as "failure case" (FC). When multiple valid solutions are found, the final solution is chosen to be the one with the lowest electrode length. The model used for evaluation is the one that yields the highest accuracy on the validation set during all the training episodes.

## 4. Experiments and Results

First, we summarise the experiments done below and discuss the corresponding results later.

(I) We evaluate the DDQN, DDQN with HER and PPO approaches.

(II) We compare the inference time, electrode length, and minimum distance of electrode to each OAR given by the DRL method (PPO) and a conventional method (Liang et al., 2019). Since we only consider single needle ablation, the implementation is simplified and does not consider multi-needle or pullback technique as in the original work.

(III) We evaluate the robustness of DRL method (PPO) with respect to tumor location since the liver can be divided into 9 segments as illustrated in Figure 3. To do so, we select a number of test subjects having tumors exclusively present in 6 or more liver segments. With this selection criteria, we get 7 test subjects and present their results in Table 3.

**Results:** (I) We evaluated all the DRL approaches for five $\gamma$ values of 0.1, 0.2, 0.3, 0.4, 0.5 and present them in Table 2. With DDQN, modeling the discrete displacement of electrode endpoint led to good results as presented in the first row in Table 2. We get a few failure cases with the lowest values observed for gamma values of 0.1, 0.4. With the addition of HER to DDQN, we do not observe significant improvements despite considering additional terminal states that satisfy a subset of constraints. Surprisingly, the continuous displacement of electrode endpoint done with proximal policy optimization (PPO) also provided comparable results as DDQN. Similarly, we get low number of failures cases, with the lowest value observed for gamma values of 0.1, 0.3. The conventional method gives an accuracy of 100% with zero failure cases.

| Method | $\gamma$=0.1 | | $\gamma$=0.2 | | $\gamma$=0.3 | | $\gamma$=0.4 | | $\gamma$=0.5 | |
|---|---|---|---|---|---|---|---|---|---|---|
| | MA | FC | MA | FC | MA | FC | MA | FC | MA | FC |
| discrete action (DDQN) | 0.986 | 2 | 0.972 | 4 | 0.972 | 4 | 0.986 | **2** | 0.972 | 4 |
| discrete action with HER | 0.979 | 3 | 0.993 | **1** | 0.972 | 4 | 0.979 | 3 | 0.979 | **3** |
| continuous action (PPO) | 0.993 | **1** | 0.972 | 4 | 0.993 | **1** | 0.979 | 3 | 0.958 | 6 |

**Table 2:** We obtain high mean accuracy (MA), and a low count of failure cases (FC) on the test set (140 test cases) for the evaluated DRL approaches on five gamma values $\gamma$.

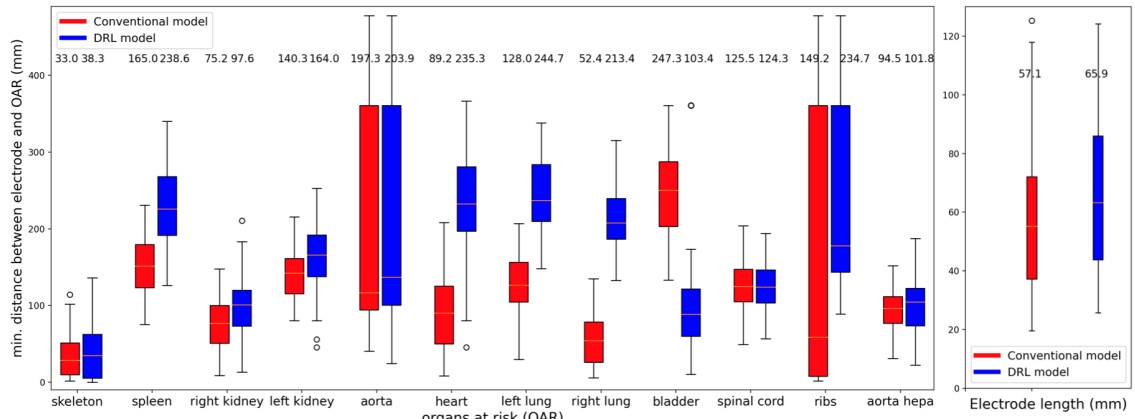

**Figure 2:** We compare results in terms of electrode length and minimum distance between the electrode and each organ at risk (OAR) for both DRL (PPO) method and re-implemented conventional method. We obtain comparable results to conventional method.

(II) Next, we compare the best DRL approach (PPO model with a gamma value of 0.1) with a simplified implementation of the conventional approach on the test set. First, we observe a mean inference time per subject of 13.3 seconds (Quadro RTX8K GPU) for the DRL approach which is 10 times faster than the conventional model that takes 135 seconds. From Figure 2, we observe that the mean distance of the electrode to each organ at risk is higher for the DRL approach except for the bladder. But, the mean electrode length is slightly higher for the DRL approach.

(III) Lastly, we evaluate if the best DRL approach (PPO) can provide an ablation plan irrespective of the tumor location within the liver (Fig. 3). With the defined selection criteria, we get 7 patients with tumors in 6 or more segments, with a total of 50 tumor cases. For this evaluation, we used the same PPO model trained for experiment (I) with $\gamma$ value being 0.1. We obtain solutions without any failure for all the tumor cases as shown in Table 3.

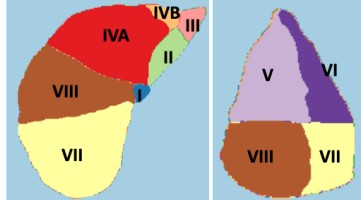

**Figure 3:** Axial, Coronal slice showing the 9 liver segments

| Segment No: | I | II | III | IVA | IVB | V | VI | VII | VIII |
|---|---|---|---|---|---|---|---|---|---|
| Tumor count | 2 | 6 | 2 | 5 | 7 | 7 | 7 | 7 | 7 |
| MA | 1 | 1 | 1 | 1 | 1 | 1 | 1 | 1 | 1 |
| FC | 0 | 0 | 0 | 0 | 0 | 0 | 0 | 0 | 0 |

**Table 3:** We present mean accuracy (MA) and failure count (FC) on 7 subjects that have tumors in 6 or more segments. We get 100% accuracy with 0 failures.

## 5. Conclusion

Manual planning of liver tumor thermal ablation is a challenging and time-consuming task. The drawbacks of previous automation works are their high inference time while supervised deep learning methods require large labeled datasets for training. We mitigate the above limitations by leveraging a deep reinforcement learning method that provides ablation plans

with low inference time and does not require labels during training. For this, we compare two popular approaches to model the electrode displacement with either discrete actions using double deep Q networks, or continuous actions using proximal policy optimization. We obtain solutions that satisfy all clinical constraints comparable to the conventional method but with 10 times faster inference time. Additionally, we demonstrate the model's robustness to provide solutions irrespective of the tumor location in the liver. We discuss the future scope of work in the appendix.

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

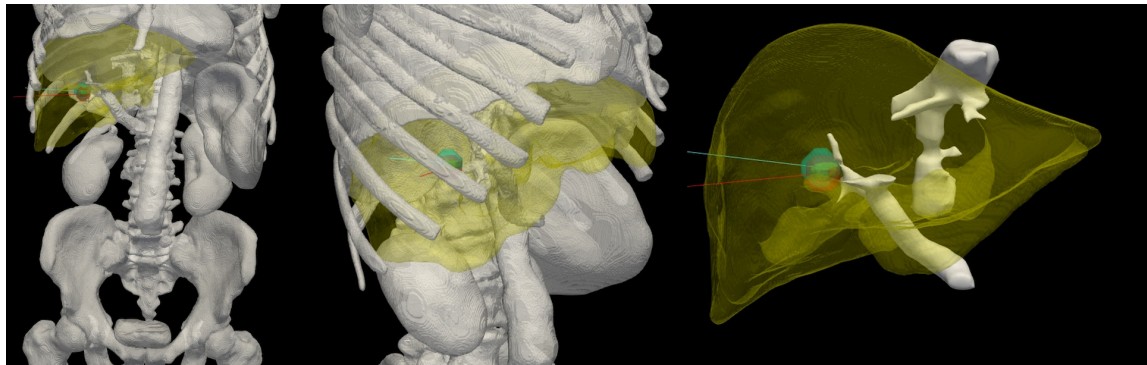

**Figure 4:** We select a tumor from a randomly chosen test subject and present the electrode trajectory results for the conventional method in blue color and DRL approach in red. The OAR and vessels are shown in white, liver in yellow, and tumor in green color.

an adversarial image-to-image network. In *International Conference on Medical Image Computing and Computer-Assisted Intervention*, pages 507–515. Springer, 2017.

Chao Yu, Jiming Liu, and Shamim Nemati. Reinforcement learning in healthcare: A survey. *arXiv preprint arXiv:1908.08796*, 2019.

Qi Zhang, Meng Li, Xiaozhi Qi, Ying Hu, Yongmei Sun, and Gang Yu. 3d path planning for anterior spinal surgery based on ct images and reinforcement learning. In *2018 IEEE International Conference on Cyborg and Bionic Systems (CBS)*, pages 317–321. IEEE, 2018.

Shangtong Zhang and Richard S Sutton. A deeper look at experience replay. *arXiv preprint arXiv:1712.01275*, 2017.

## 6. Appendix

**Future scope of work:** (1) We aim to include an evaluation by clinical experts of the proposed solutions from DRL and conventional methods to establish if they are acceptable or not as ground-truth trajectories are not provided in the dataset. (2) We plan to address the complex scenarios of multi-needle and big tumors ablation with pullback technique with a single electrode.

In practice, clinicians base their planning on one CT scan even though multiple scans are acquire during the procedure. The proposed method could lead to better planning by re-evaluating plans quickly for each new CT scan.

