# OpenReview forum: "Automatic planning of liver tumor thermal ablation using deep reinforcement learning"
_MIDL.io/2022/Conference — MIDL 2022_

### Official Review · Reviewer_9eBc · 2022-01-21

**Confidence:** 5
**Preliminary Rating:** 5
**Recommendation:** Poster

**Summary:**

The paper proposed to use RL for automated planning of thermal liver tumor ablation. Robust results for needle trajectory planning are shown for the liver tumour segmentation (LITs) challenge dataset. Resuls are robust with better placement resutls than SOTA and 10x faster computation speeds. The method is evaluated on public data and thus reproducable.

**Strengths:**

This is a well executed paper with an interesting application of reinforcement learning to liver tumor ablation planning.
The approach is 10x faster than the state-of-the-art and yields likely significantly better results in terms of placement accuracy.


**Weaknesses:**

- not all relevant structures have been considered
- the procedure is often more complex than modelled (only single electrode has been modelled)
- reproduction of the planned intervention has not been discussed


- minor: some parts are redundant like the repeated emphasis that this task is very 'tedious'. The paper could be streamlined a bit.


**Deanonymize Review:**

no

**Detailed Comments:**

there is more to avoid than hepatic vessels. There are arteries, portal vein etc. The model might be too simplistic to calculate optimal trajectories, especially because the placement close to larger vessels will often cause cooling and thus insufficient ablation.

thermal ablation is usually done with either several ablations in case of single electrodes or several electrodes at once or special electrodes that fan out. This has not been modelled and should be discussed.

How would the proposed trajectory reproduced by an interventional radiologist in practice?

**Final Rating After The Rebuttal:**

5: Strong Accept

**Justification Of The Final Rating:**

Most of my comments were re future work and these have been acknoleged. This is an interesting paper.
Thank you for addressing the points raised above and adding clarifications to the presented work. I have no further comments.

**Paper Type:**

methodological development

**Questions To Address In The Rebuttal:**


How would the proposed trajectory reproduced by an interventional radiologist in practice?

Could the placement be optimised in terms of treatment outcome/success instead of only spatial placement mainly depending on avoiding critical organs. I guess so, but please comment and discuss in the paper.

**Special Issue:**

no

---

### Official Review · Reviewer_AVuy · 2022-01-22

**Confidence:** 4
**Preliminary Rating:** 3
**Recommendation:** Poster

**Summary:**

The paper at hand addresses the problem of trajectory planning for minimally invasive liver tumor ablation. The authors propose to use a deep reinforcement learning approach using clinical constraints as a reward. The authors compare to a conventional optimization method, reporting similar performance while reducing the planning time by a factor of 10. A public liver tumor segmentation dataset is used.

**Strengths:**

•	The paper is well-written and a pleasant read

•	The choice of DRL appears reasonable for the problem at hand

•	The authors use data from a public source

•	The authors report comparable performance to a conventional method while being much faster


**Weaknesses:**

•	Novelty is limited as the paper present the use of existing methods on a different problem

•	While the method description is detailed, the results are a bit too short, e.g., there is no visualization of resulting paths

•	The authors’ approach is only compared to one other method. It also appears to be a modified re-implementation without much detail provided which makes the results hard for the reader to judge

•	The authors only evaluate failure rates and distance to OARs – there is no actual evaluation using ground-truth trajectories by experts or an assessment of the resulting trajectories by experts

•	While inference time is mentioned, training times are not addressed.

•	Results from Fig.3 appear somewhat cherry-picked


**Deanonymize Review:**

yes

**Detailed Comments:**

In general, the paper is interesting but there are several points that should be addressed.

Novelty:

Since this is an application paper, it is okay for the novelty to be limited. However, it should be noted that reinforcement learning for surgery planning has been investigated for many applications (as cited by the authors). Therefore, there is not much differentiating the authors’ work from others’.

Results:

The amount of results is limited. In particular, I would have liked to see some visualizations of the resulting paths. Furthermore, comparisons to more conventional/other methods would be helpful. Also, details on the modification of the one conventional method would be appreciated.

Evaluation:

While demonstrating that the method does not violate the set boundaries and often reaches the target, the authors do not appear to compare the methods to actual ground-truth trajectories, assumingly, due to lack of availability. However, this limits the reader’s ability to judge the methods’ performance/suitability for the task. Alternatively, an expert’s evaluation of some resulting trajectories could have been an alternative.

Training times:

One of the big downsides of RL is the very long training. The authors should report and comment on training times.

Fig.3:

Showing results from just 5 patients could be interpreted as cherry-picking. Why not show the results per liver region across the entire test set?


**Final Rating After The Rebuttal:**

4: Weak Accept

**Justification Of The Final Rating:**

The authors addressed all my comments and gave mostly adequate explanations. Requested modifications to the manuscript were made (plotting a trajectory). However, my point regarding evaluation against actual ground-truth trajectories or evaluation by clinical experts remains (which is understandable given a GT would be hard to obtain). Thus, I am changing my rating from borderline to weak accept.

**Paper Type:**

validation/application paper

**Questions To Address In The Rebuttal:**

See weaknesses. I understand that space in the paper is limited but the weaknesses should at least be shortly addressed. The method description could be shortened instead. Results and evaluation are most important to me.

**Special Issue:**

no

---

### Official Review · Reviewer_nxAy · 2022-01-22

**Confidence:** 5
**Preliminary Rating:** 1
**Recommendation:** Poster

**Summary:**

The authors propose a deep reinforcement learning (DRL)-based method to plan thermal ablation procedures of tumors in the liver. They motivate the use of DRL with low inference time compared to existing optimization methods. They consider three DRL algorithms: DDQN, DDQN + HER and PPO. They train and test their method on public dataset of annotated liver tissues and tumors. Their methods exhibit high accuracy on the considered dataset. They also compare themselves to a conventional method and show that their solution is generally further from organs at risk than the conventional method. Finally, the authors evaluate the robustness of their trained PPO algorithm w.r.t tumor location on the liver and exhibit no failure cases this time, and perfect accuracy.

**Strengths:**

DRL, compared to supervised learning, has many more "moving parts" and the authors did well on describing each one of them. The environment, state space, action space, reward and episode modalities are well described. The authors also considered two variants of their environment (discrete and continuous) with corresponding algorithms and explored multiple discount factors. The proposed method does answer to the main motivation, which is to reduce inference time.

**Weaknesses:**

However, the main contribution of the method, which is the inference time, seems poorly motivated. While the proposed approach is ten times faster than the "simplified conventional approach", the conventional approach only takes around two minutes to plan the thermal ablation procedure, which is not a lot if it can guarantee a proper ablation procedure. Does the procedure need to be recalculated many times ? Or even in real-time ? If so, then I agree that the proposed DRL method is superior (even so, 13 seconds is still marginally slower than real-time). Otherwise, waiting two minutes once does not seem too bad, no ? Generally, the problem and motivations need to be better defined to justify the method.

Moreover, some aspects of the experiment are confusing: it is mention a few times that the conventional method was "simplified" but it is unclear how, why and what the impacts of this simplification are. The accuracy and failure cases of the conventional method are also not considered in experiment 1. Is it because the conventional method is always 100% accurate and does not cause failure cases ? If so, it should be mentioned in the text. Especially when considering the possibly marginal time difference between the DRL and conventional approaches, two minutes of waiting time might be a small price to pay for correctness guarantees.

The gamma values considered in experiment 1 are also "out of the ordinary". Robotic control and game-playing, which are the standard problems for modern deep reinforcement learning, usually consider discount factors around 0.9-0.99 [1, 2]. As it stands, a discount factor of 0.1 renders any reward past the one at time $t$ meaningless and makes the agent almost purely greedy. This should be discussed.

The differences between the first and third experiments are also unclear. Why is it that PPO encounters failure cases and non-perfect accuracy during the first experiment but not the third ? Why was DDQN (+ HER) not considered for this experiment ? Is it because it did not reach the same perfect-level of performance as PPO on this experiment ? This experiment is seriously lacking details.

Finally, the text has a few confusing parts that are addressed below and should be corrected.

[1] Mnih, V., Kavukcuoglu, K., Silver, D., Rusu, A. A., Veness, J., Bellemare, M. G., ... & Hassabis, D. (2015). Human-level control through deep reinforcement learning. nature, 518(7540), 529-533.

[2] Schulman, J., Wolski, F., Dhariwal, P., Radford, A., & Klimov, O. (2017). Proximal policy optimization algorithms. arXiv preprint arXiv:1707.06347.

**Deanonymize Review:**

no

**Detailed Comments:**

- Introduction, page 1: "Liver cancer is the fourth highest occurring cancer type"

Where ? In the world ? The sentence should mention.

- Section 1, page 2: "Also, DRL approaches do not involve any processing steps as in conventional methods and can generalize well at test-time due to the nature of training."

This should be included in the previous paragraph as this pertains to general DRL approaches, not only to the one proposed by the authors.

- Section 1, page 2: "we explore two approaches: (a) in the first one, we explore a discrete displacement of electrode endpoint with a double deep Q networks (Van Hasselt et al., 2016) (DDQN) that estimates discrete action values. (b) in the second ... "

HER should be mentioned here as well.

- Section 1, page 2: "We also obtain clinically acceptable solutions irrespective of the tumor location within the liver in any of the 9 segments, indicating the robustness of the proposed approach."

This sentence needs clarification or rewording to explain what the nine segments are and what these results entail.

- Section 2.1, page 3: "In Fig. 1.a., we present the custom environment containing 3D information of the tumor, organs at risk, skin masks denoted by m"

Does $m$ include all the volumetric information ? The content of $m$ should be explicitly defined.

- Section 2.1 page 3-4:

subsections 2.2a, 2.2b should be 2.1a, 2.1b instead.

- Section 2.1 page 4: "We stop the episode at the step when all constraints are satisfied, and this state is called terminal or final state."

This was already previously mentioned in page 3.

- Section 2.2 page 4: "the dense layer output 27 $(3 \times 3)$ Q-values"

This should be $(3 \times 3 \times 3)$, no ?

- Section 3 page 6:

the steps in "pre-processing" change tone quite a bit: starts passive ("dilation of 1mm"), then becomes "personnal" ("we re-sample"). The final step is also not numbered ("Finally, we crop"). The text should be rewritten to be consistent.

- Section 3 page 7: "If one or more valid solutions are found, the accuracy is set to 1 else to 0 (failure case)."

The sentence should be re-written or split to be clearer: "If a valid solution is found, the accuracy is set to 1. Else, it is set to 0 and is declared at "failure case" (FC). When multiple ..."

- Section 4 page 7:

The enumeration of the experiment should be rewritten as full sentences forming a paragraph or multiple to be clearer. Moreso, the first experiment should not be split into three parts corresponding to each algorithm. Finally, the dataset used should be recalled in this section. Experiment 2 is poorly described and the corresponding sentence should be rephrased. Experiment 3 would also benefit having much more details, which would shed light on the discrepancies in results between experiment 1 and 3.

- Section 4 table 2:

The best results in each column and row should be put in bold to help the reader assess which method performs best.

- Section 4 page 8:  "From Table 2, we observe that the mean distance of the electrode to each organ at risk"

The text should point to "Figure 2".

- Section 5 page 8: "Solutions are obtained for all the selected tumors irrespective of their locations in the liver segments for all patients, demonstrating the robustness of the DRL approach."

The discrepancies between theses results and the results from experiment 1 should be mentioned and discussed.

**Final Rating After The Rebuttal:**

4: Weak Accept

**Justification Of The Final Rating:**

The authors did well on justifying several points raised in the initial review as well as adding clarifications to the presented work. The authors have greatly improved the quality of the presented work through the revision process and the paper is now, in my opinion, suitable for MIDL.

**Paper Type:**

methodological development

**Questions To Address In The Rebuttal:**

The problems raised in the "weaknesses" section, and the errors or problems in the text raised in the "detailed comments", should be addressed. Importantly, the authors should mention if the planning procedure has to be done more than once or even possibly in real-time, both in the rebuttal and in the text, to truly motivate their method. The authors should also mention the accuracy and failure cases of the conventional method, as well as provide more details on it.

**Special Issue:**

no

---

### Meta-Review · Area_Chair_YLqS · 2022-02-18

**Recommendation:** Accept (Poster)
**Confidence:** 5

**Metareview:**

The work describes the use of RL for trajectory planning for minimally invasive liver tumor ablation. The main motivation is that RL can accelerate planning over conventional methods.

Strengths according to reviewers:
* Interesting problem.
* Relatively well written.
* Good description of RL methodology.
* Interesting results (comparable performance to conventional method, faster inference)
* Uses public database, which may facilitate reproducibility and comparisons in future work.


Weaknesses:
* Limited novelty with respect to machine learning. Main contribution is rather the idea of using RL to improve the specific application.
* Not extensive evaluation.

Initial reviewer concerns regarding weak motivation and clarity were addressed to a significant extent, improving the manuscript, which is reflected to the increased ratings after rebuttal.

After a very productive rebuttal and discussion period, accompanied with significant updates to the paper, there seems to be a concensus that the paper is of acceptable quality for a publication in MIDL.

---

### Decision · Program_Chairs · 2022-02-28

Accept